# Energy and Exergy Analyses of Rice Drying in a Novel Electric Stationary Bed Grain-Drying System with Internal Circulation of the Drying Medium

**DOI:** 10.3390/foods11010101

**Published:** 2021-12-31

**Authors:** Guiying Wang, Wenfu Wu, Daping Fu, Wen Xu, Yan Xu, Yaqiu Zhang

**Affiliations:** 1College of Biological and Agricultural Engineering, Jilin University, 5988 Renmin Road, Changchun 130025, China; guiying9602@163.com (G.W.); fudaping@jlau.edu.cn (D.F.); xuyan@jlu.edu.cn (Y.X.); yaqiu@jlu.edu.cn (Y.Z.); 2Jilin Business and Technology College, 1666 Cullen Lake Road, Changchun 130507, China; xuwen2004@sina.com

**Keywords:** internal circulation of the drying medium, electric power, rice drying, energy and exergy analyses

## Abstract

In our study, we developed a system to reduce both energy consumption and pollutant discharge during the drying process. We present a new technology, a stationary bed grain-drying test device based on the internal circulation of the drying medium (ICODM). A rice-drying experiment was carried out inside of it, and the influences of air temperature (AT) and air velocity (AV) on the energy and exergy efficiencies (EEE) as well as the improvement potential rate (IPR) and the sustainability index (SI) of the rice-drying process were studied. The following conclusions were obtained: when the rice was dried at a temperature of below 55 °C and an AV across the grain layer of 0.5 m/s, the average EEE during the drying process was 48.27–72.17% and 40.27–71.07%, respectively, demonstrating an increasing trend as the drying medium temperature increased. When the rice was dried using an AV across the grain layer in the range of 0.33–0.5 m/s and a temperature of 40 °C, the two values were 39.79–73.9% and 49.66–71.04%, respectively, demonstrating a decreasing trend as the drying medium flow velocity increased. IPR and SI were 4.1–8.5 J/s and 1.9–2.7, respectively, at a drying temperature of 30–55 °C and an AV of 0.33–0.5 m/s. These conclusions can provide helpful guidance for the optimization and control of the rice-drying process in terms of saving energy.

## 1. Introduction

Drying is an operation that involves high energy consumption and pollution, and drying operations consume approximately 10–15% of the total national industrial energy consumed in Canada, France and the USA, and 20–25% of that in Germany and Denmark [1]. In China, drying operations account for 12% of the total energy consumption of the national economy [2]. Against the background of the global fossil energy crisis and environmental pollution, energy savings and environmental protection are goals being pursued by the drying industry; given the increase in grain output and the acceleration of agricultural mechanization in China, the development of energy-saving methods and environmentally-friendly grain drying is an urgent requirement for agricultural development. Researchers are seeking ways to improve the energy efficiency of dryers through such structural design elements as improving the thermal efficiency of heat sources and improving the heat preservation of the hot blast stove, air duct, and dryer exhaust air recirculation system [3]. In the pursuit of greater energy savings and environmentally-friendly hot blast stoves from dry heat sources, coal-fired, fuel gas, and biomass-based hot blast stoves, heat pumps, solar energy, infrared drying, and dryers based on the combination of several heat sources have all been developed [4,5,6,7,8]. Electric drying is seldom used in large and medium-sized grain dryers because of its high power configuration; however, electric energy storage-type hot blast stoves for grain drying have emerged in recent years as a new way to utilize electrical energy for drying via energy storage [9]. The drying device developed in this paper uses clean electrical energy as a drying heat source, and the exhaust air is recycled to form a closed loop to achieve the purposes of saving energy and reducing the power configuration of the dryer [10,11,12].

Structurally, rice consists of a hard husk and seeds. When it is dried, the outer shell blocks the transfer of water from the inside of the grain to the outer surface; therefore, rice is a difficult grain to dry. In addition, rice is sensitive to high heat, and the maximal drying temperature and drying time should be strictly controlled during the drying process to ensure its post-drying quality [13]. The electric heating adopted by the device has the advantages of precise temperature control and small lag, which help to ensure optimal drying process parameters.

In recent years, exergy analysis based on the conservation of mass and energy and the second law of thermodynamics has attracted increasing global interest from scholars. Exergy analysis is used in the drying process to evaluate dryer performance through grain-drying experiments on different types of dryers, during which the location and amount of exergy loss can be found in order to improve drying efficiency and reduce the impact on the environment. Li et al. [14] from South China Agricultural University evaluated the performance of a combined infrared radiation–counterflow circulation corn dryer on the basis of energy efficiency, heat loss, exergy flow, and exergy efficiency. Silva et al. [15]. analyzed the exergy characteristics of a solar dryer in a mixed chamber through a corn drying test, and concluded that thermal and exergy efficiencies show opposing trends as the drying process proceeds. Li et al. [16] from South China Agricultural University conducted exergy economic analysis on an industrial maize drying system, and reached the important conclusion that the order of exergy loss impacts the total exergy loss of the main components of the drying system, and found that the exergy efficiency of the drying system varied in the range of 14.81–40.10%. Wincy et al. [17] demonstrated the exergetic evaluation of a biomass gasifier-operated reversible flatbed dryer for paddy drying in the parboiling process, obtained the proportion of each component in the total exergy of the system, and concluded that the average exergy efficiency of the overall drying system was 21.28%. Islam et al. [18] studied the energy, exergy, and milling performance of parboiled paddy in an industrial LSU dryer, and found that the change rule of exergy inflow, exergy losses, and exergy efficiency of the dryers was 43.63–67.21%. Zohrabi et al. [19] studied the exergetic performance of a pilot-scale convective dryer with exhaust air recirculation by a wood-chip drying experiment, and found that increasing the drying temperature, air volume flow rate, and recycling fraction improved the overall functional exergetic efficiency of the drying system, while exhaust air recirculation profoundly improved the overall functional exergetic efficiency of the drying system. Exergy analysis has been an important analytical method for scholars seeking to optimize control and improve the energy efficiency of drying systems over the past three decades [20,21]. Because the energy consumption index is jointly determined by dryer performance and drying process optimization, for some grain-drying systems the optimization of the drying process plays an important role in the evaluation of the energy-saving index. In recent years, exergy analysis has been used to analyze the effect of drying process parameters on exergy efficiency, to optimize the drying process, and to increase the energy efficiency of drying systems. Studies in this area include Khanali et al. [22], who studied the effect of inlet drying air temperature, feed mass flow rate and weir height on the exergetic performance of a plug flow fluidized bed drying process for rough rice. Beigi et al. [23] studied the effect of exergetic performance on deep-bed drying of rough rice in a convective dryer. Yogendrasasidhar, D. and Setty, Y.P. [24] studied the effect of wall temperature, air velocity and bed height on the exergy efficiency of Kodo millet grain and fenugreek seed drying in a wall-heated fluidized bed dryer. Pattanayak et al. [25] studied the effect of drying air temperature and paddy mass in a batch on exergy performance in a vertical fluidized bed dryer. Sarker et al. [26] studied the effect of drying temperature and initial moisture on the energy and exergy performance of paddy drying in an industrial fluidized bed. Tohidi, M. et al. [1] studied the effect of the temperature, relative humidity, and flow rate of drying air on energy efficiency in a fixed deep bed dryer. The research conclusions drawn from these studies are shown in Table 1. Through these studies, have scholars determined the law governing the influence of the drying process parameters of different types of grain dryers on exergy efficiency.

In order to design a high-efficiency and energy-saving grain dryer, we propose using an internal circulation grain-drying system in which the entire exhaust of the dryer is recovered and the drying medium forms a closed loop in the system, which could save a significant amount of heat energy. Static bed drying is a traditional form of rice drying; the new idea for grain drying based on ICODM is used along with static bed drying. The method of drying through flow is adopted, where the heat source is electric energy [27]. The stationary bed grain-drying test device represents a novel design, and was built and used to study the influence of drying process parameters on exergy efficiency in the drying process and to provide guidance for the development of optimized grain-drying processes.

In this paper, a rice-drying experiment was carried out in a stationary bed grain-drying apparatus with ICODM. The influences of AT and AV on the EEE, IPR, and SI of the drying process were analyzed. This study offers support for improving the design of drying systems, the setting of technological parameters, and the development of control systems. Exergy analysis of a stationary bed drying system for rice based on ICODM has, to the best of the authors’ knowledge, not previously been conducted.

## 2. Materials and Methods

### 2.1. Materials and Experiment Conditions

The rice used in the experiment was newly harvested Dragon rice 16 from Jilin province in China, which belongs to the sweet rice variety. Samples were harvested from the ground in 2 kg bags, sealed in plastic bags, and kept fresh in refrigerators. Initial moisture was about 27%, which was measured using a PM-8188 grain moisture meter produced by KETT Japan. During the test, samples were taken out and placed in the environment for about 10 min. Then, 340 g of the sample was weighed for the drying test. The weight and moisture of the samples were measured before and at the end of the test.

As low-temperature drying is conducive to maintaining the quality of rice [25,28], six temperature levels, namely 30, 35, 40, 45, 50, and 55 °C, were selected to analyze the exergy characteristics of the drying chamber during the rice-drying test. For the variable air temperature experiment, the air velocity across the grain layer was 0.5 m/s; for the variable air velocity experiment, the temperature was 40 °C.

### 2.2. Experimental Apparatus

The experimental apparatus was designed according to the principle of the ICODM. It belongs to the category of flow tray drying of a stationary bed drying in drying mode. Because the hot air flowing through the material layer is turbulent, the lagging layer of airflow in the material layer on the particle surface is thin, such that the heat conduction of hot air into convection heat transfer can improve the heat transfer coefficient. At the same time, the utilization of the heat carrier is improved; thus, energy consumption is reduced [27] and energy efficiency is further improved through ICODM. The development of the whole device is greatly significant for saving energy, shortening the production cycle, and improving product quality. It can also provide an experimental reference for the development of large grain dryers based on ICODM. Photographs of the device are shown in Figure 1, and the design is shown in Figure 2. The experimental apparatus consists of a drying chamber, a water-cooled condenser, an electric heater, a fan, a detection sensor, and a control host. The drying chamber, which is the main component of grain drying, stores wet grain that needs to be dried. The water-cooling condenser is used to condense the hot and humid exhaust gas discharged from the drying chamber, and cooling is provided by the flow of cold water through the water-cooled plate. The drying medium flowing through the electric heater is heated using the heat that is provided by an electric heater wire of appropriate power in an electric heater. The fan provides the power to circulate the drying medium through the system, making the drying medium form a closed loop in the system. Fan speed can be adjusted via the power supply voltage, and features a pulse speed measurement function. The detection sensor includes a hot air temperature sensor, a condenser inlet and outlet water temperature sensor, and a temperature and humidity sensor before and after exhaust condensation. The control host controls the starting and stopping of all system components, monitoring, and data storage. LabVIEW 2018 software was used to develop the window display control interface and to set the drying process parameters and the real-time display interface.

Table 2 lists the specifications, models and parameters of the main components of the system; Table 3 lists the measurement parameters and detection components.

### 2.3. Energy and Exergy Analysis Principle

In the electric stationary bed grain-drying system with ICODM, the drying medium forms a closed loop in the device. The grain remains stationary throughout the drying process, circulating flow through the drying medium inside the device and constantly removing water from the grain. As the drying process progressed, the water in the drying medium absorbed from the grain was not fully condensed by the condenser; its relative humidity showed a trend of gradual increase. The energy conversion and distribution in the device are shown in Figure 3. The following were assumed for exergy analysis in this paper:(1)The drying medium and grains suffered no losses through leakage;(2)When the drying medium flowed inside the device, energy loss between components was included in total loss;(3)Changes in the kinetic energy and pressure of the drying medium were ignored in exergy analysis;(4)The initial grain temperature was regarded to be consistent with room temperature, and the temperature in the drying process was regarded as consistent with the hot air temperature.

Exergy is a relative quantity based on a given environment. For the exergy analysis of a working process in a test system, the first step is to determine the exergy reference point, which is the environmental state [29]. The main components of the testing device were located inside the outer box; thus, the state of the inner part of the outer box was the reference for the exergy analysis.

#### 2.3.1. Energy Analysis

According to the first law of thermodynamics, the total energy of a system is conserved. According to the analysis in Figure 3, the following equation was obtained:(1)$$\dot{W}={\dot{Q}}_{e}+{\dot{Q}}_{a1}+{\dot{Q}}_{g}+\sum {\dot{Q}}_{loss}$$
where $\dot{W}$ is the total electrical energy provided by the electric heater, which was measured by the power meter; ${\dot{Q}}_{e}$ is the heat required by the evaporation of grain water, which belongs to the useful energy of the system; ${\dot{Q}}_{a1}$ is the heat required to heat up the drying medium; ${\dot{Q}}_{g}$ is the heat required to heat up the grain; $\sum {\dot{Q}}_{loss}$ is the total heat loss of the device, according to the heat balance equation of the drying medium and grain in the drying chamber; and ${\dot{Q}}_{e}$ was derived and calculated using the following equation [30]:(2)$${\dot{Q}}_{e}=n\times \left(2500+1.842t_{a2}-C_{w}\theta_{g1}\right)$$
where *n* is a constant and denotes water removal per unit time. The energy efficiency of the drying process was calculated by the following equation:(3)$$\eta =\frac{{\dot{Q}}_{e}}{{\dot{Q}}_{a1}}$$
where ${\dot{Q}}_{a1}$ was calculated using the following equation:(4)$${\dot{Q}}_{a1}={\dot{m}}_{a}C_{a}\left(t_{a1}-t_{a3}\right)$$

#### 2.3.2. Exergy Analysis

When the exergy analysis of the drying process was carried out, the exergy of the drying medium and grain was expressed in the form of exergy rate according to the basic state parameters [20]:(5)$${\dot{E}}_{\chi a}={\dot{m}}_{a}\left\{\begin{array}{l} \left(C_{a}+\varphi C_{v}\right)\left(t-t_{0}\right)-t_{0}\left[\left(C_{a}+\varphi C_{v}\right)\ln\left(\frac{t}{t_{0}}\right)-\left(R_{a}+\varphi R_{v}\right)\ln\left(\frac{P}{P_{0}}\right)\right]\\ +t_{0}\left[\left(R_{a}+\varphi R_{v}\right)\ln\left(\frac{1+1.6078\varphi_{0}}{1+1.6078\varphi }\right)+1.6078\varphi R_{0}\ln\left(\frac{\varphi }{\varphi_{0}}\right)\right] \end{array}\right\}$$
(6)$${\dot{E}}_{\chi g}={\dot{m}}_{g}C_{g}\left[\left(t-t_{0}\right)-t_{0}\ln\frac{t}{t_{0}}\right]$$

The exergy balance equation used for the drying process was developed in the following rate form [20]:(7)$${\dot{E}}_{\chi g2}-{\dot{E}}_{\chi g1}={\dot{E}}_{\chi a1}-{\dot{E}}_{\chi a2}+{\dot{E}}_{\chi e}-{\dot{E}}_{\chi l}-{\dot{E}}_{\chi des}$$
where ${\dot{E}}_{\chi e}$ is the exergy rate for the water evaporating from the grain, which was calculated following [20]:(8)$${\dot{E}}_{\chi e}=\left(1-\frac{t_{0}}{t_{g}}\right){\dot{Q}}_{e}$$

${\dot{E}}_{\chi l}$ is the exergy loss caused by the thermal loss when exergy flowed through the test device, which was calculated following [20]:(9)$${\dot{E}}_{\chi l}=\left(1-\frac{t_{0}}{t}\right){\dot{Q}}_{l}$$
(10)$${\dot{Q}}_{l}={\dot{Q}}_{e}+{\dot{m}}_{a}\left(h_{a1}-h_{a2}\right)+{\dot{m}}_{g}\left(h_{g1}-h_{g2}\right)$$

This heat loss could be reduced by the installation of insulation.

Exergy destruction is also called inevitable exergy loss, which was calculated following [21]:(11)$${\dot{E}}_{\chi des}=t_{0}{\dot{S}}_{gen}$$

The exergy efficiency of the drying process was determined as follows:(12)$$\psi =\frac{{\dot{E}}_{\chi e}}{{\dot{E}}_{\chi a1}}$$

The IPR of the drying process was calculated following [19]:(13)$$I\dot{P}=\left(1-\psi \right)\left({\dot{E}}_{\chi in}-{\dot{E}}_{\chi out}\right)$$
(14)$${\dot{E}}_{\chi in}={\dot{E}}_{\chi g1}+{\dot{E}}_{\chi a1}$$
(15)$${\dot{E}}_{\chi out}={\dot{E}}_{\chi g2}+{\dot{E}}_{\chi a2}$$

The SI of the drying process was calculated following [19]:(16)$$SI=\frac{1}{1-\psi }$$

## 3. Results and Discussion

Because the drying energy consumption index is jointly decided by drying machine performance and drying process, analysis of the influences of the drying process parameters on the exergy efficiency of the drying process can provide an important reference for the optimization of the drying process. The effects of AT and AV on the exergy efficiency of the drying process are analyzed in this paper.

### 3.1. Effect of Drying-Medium Temperature on Exergy Efficiency of Drying Process

Figure 4 shows that the energy efficiency changed over time in the rice drying process. When the rice was dried at 30, 35, 40, and 45 °C, the energy efficiency of the drying process gradually improved as the drying process progressed. This was due to the use of circulating water as the condensation medium in the testing process. As the test progressed, the water temperature of the condensing medium used for the condenser gradually rose, the condensation effect weakened, and the temperature of the drying medium after condensation rose, such that the drying inlet energy decreased and the amount of energy required to evaporate a kilogram of water barely changed, leading to a gradual increase in drying energy efficiency. When drying at 50 and 55 °C, the temperature difference between hot and cold fluids was large, and the condensation effect was good. The drying process was little affected by the increase in circulating water temperature; thus, there was little change in energy efficiency during the drying process.

The curve of exergy efficiency change with drying time in Figure 5 shows that the exergy efficiency of the rice dried at 30, 35, 40, and 45 °C changed little over time, with a fluctuation range of 3.3–4.7%, while exergy efficiency decreased gradually when the rice was dried at 50 and 55 °C. This is because in the drying process, the temperature of the grain and drying medium gradually increased, and the exergy rate of the two also gradually increased, while the exergy rate for grain moisture evaporation barely changed; thus, the exergy efficiency gradually decreased. When drying using temperatures below 50 °C, the temperature difference between grain and drying medium in the reference state was small, and the temperature change of the two had little effect on the exergy rate during the drying process; thus, the effect on the exergy rate was not obvious. The exergy efficiency of the drying process was 40.27–71.07% under experimental conditions, much higher than in previous results. The exergy efficiency of a plug flow fluidized bed drying process for rough rice was 4.18–12.00% in the studies of Khanali et al. [22]; the exergy efficiency of deep bed drying of rough rice in a convective dryer was 5.10–29.41% in the studies of Beigi et al [23]; the exergy efficiency of industrial LSU drying of parboiled paddy was 43.63–67.21% in the studies of Islam et al [18]; and the average exergy efficiency of a biomass gasifier-operated reversible flatbed dryer for paddy drying using a parboiling process was 21.28% in the studies of Wincy et al. [17]. The exergy efficiency of the drying process could be significantly improved with internal circulation technology for the drying medium. In accordance with the findings of Zohrabi et al. [19], exhaust air recirculation profoundly improved the overall functional exergetic efficiency of the drying system as a decision-making parameter, by as much as two times.

The curve in Figure 6 shows that the variation rule of average energy efficiencies in the drying process is not obvious, because energy efficiency is affected by both drying rate and condensation intensity. Under the same external environmental testing conditions, water temperature as a cold fluid was the same, a higher drying air temperature was adopted, and higher-temperature exhaust was discharged; then, as it had a high dew-point temperature, condensation occurred earlier when cold fluid was encountered than when a lower drying air temperature was used, meaning that more water was condensed and greater condensation intensity was obtained, and more heat was necessary to warm it to the air temperature needed for drying; thus, the drying inlet energy was higher, and the higher the drying air temperature, the higher the resulting drying rate and water evaporation energy unit time are; therefore, the energy efficiency Equation (3) shows that as the AT changes, the comparison of drying energy efficiency depends on whether condensation intensity or drying rate is dominant. In the test, apart from the fact that drying energy efficiency at 55 °C was significantly higher than at other temperature conditions, the influence of AT on energy efficiency followed no obvious rule. This is different from the research results of Tohidi et al. [1], which showed that energy efficiency increased with increasing temperature, which was because of the effect of exhaust air recirculation and condensation processes.

Exergy efficiency gradually increased with the increase in the drying AT of the drying process. This is the same as the research results of Pattanayak and Mohapatra [25], which showed that an increase in drying temperature in turn increases exergetic efficiency, as well as the research results of Zohrabi et al. [19], which showed that increasing the drying temperature improved the overall functional exergetic efficiency of drying system; in particular, it increased more obviously when the drying temperature exceeded 50 °C. Therefore, according to the principle of not affecting drying quality, a higher drying medium temperature saves more energy.

### 3.2. Effect of Air Velocity on Exergy Characteristics of Drying Process

AV is another important factor affecting grain-drying characteristics. In this paper, six AV levels were selected for the rice-drying experiment through the grain layer. The changes in the EEE of the drying process are shown in Figure 7 and Figure 8.

Figure 7 and Figure 8 show that the EEE first decreased, then increased in the drying process. This was due to the initial drying; as the drying medium temperature gradually rose, the condensation process was gradually enhanced, resulting in greater heat loss from the system. Once the drying process had proceeded for a period of time, and the condensation process was no longer enhanced, the difference between exhaust and hot air temperature after condensation no longer increased. The temperature of the outer box, where the drying chamber was located, gradually increased, while the energy required to heat the exhaust air temperature after condensation to the preset hot AT gradually decreased; therefore, the EEE increased during the drying process. The EEE was 43.7–79.5% and 59.1–83.1% at an AV of 0.3–0.43 m/s through the grain layer, respectively; it was 47.6–56.9% and 44.7–47.0% at an AV of 0.47–0.5 m/s, respectively. The value was lower than at low AV; however, with the change in drying time, it was more moderate. As the interval of AV was relatively small, the EEE values were close at AVs of 0.33, 0.37, and 0.43 m/s; overall, the higher the AV was, the lower the corresponding energy efficiency was. This is the same as the research results of Tohidi et al. [1], which showed that energy efficiency increased with a decreasing flow rate of drying air.

Figure 9 shows that the average energy efficiency of the drying process decreased with the increase in AV, while the average exergy efficiency increased slightly with the increase in air velocity at 0.33–0.4 m/s and decreased when the air velocity exceeded 0.4 m/s. Therefore, when the drying process parameters were set, the average exergy efficiency of the drying process was high when the AV passing through the grain layer was lower than 0.4 m/s.

### 3.3. Effect of Temperature of the Drying Medium on the Improvement Potential Rate and Sustainability Index of the Drying Process

Figure 10 shows that the IPR of the rice-drying process at 30, 35, 40, and 45 °C gradually decreased with the progression of the drying process, and the fluctuation range of the improvement potential rate was 4.1–8.5 J/s. This fluctuated greatly at 45 °C. When the rice was dried at 50 and 55 °C, the IPR increased gradually as the drying processed. Taking 45 °C as the cut-off point, the IPR of the drying process showed an opposite trend to drying time, mainly because the condensation effect is more obvious when the temperature is high and the heat and mass exchanger is faster during the drying and condensation process. This shows that the higher the drying AT is, the more measures such as good external insulation need to be taken to reduce the heat loss in the drying process in order to improve the test equipment and increase energy efficiency. With the increase in drying temperature, the IPR has no obvious rule, which is different from the research results of that Khanali et al. [22], which showed that the IPR of a plug flow fluidized bed drying process of rough rice increases with drying AT, mainly because exhaust air recirculation and condensation were involved and the energy exchange of drying system was more complicated.

Figure 11 shows that the SI of the rice-drying process did not change appreciably at 30, 35, 40 and 45 °C, with a value generally between 1.9 and 2.7. When drying at 55 °C, the SI gradually decreased as drying progressed until it approached the SI at a lower drying temperature. With the change in drying AT, the SI had no obvious rule. This is different from the research results of Mohsen Begi et al., which showed that the SI of the deep bed drying of rough rice in a convective dryer increased with the increase in drying AT. However, this internal circulation drying system has a higher SI with less of an impact on the environment.

### 3.4. Effect of AV on the IPR and SI of the Drying Process

As shown in Figure 12, the IPR of the drying process increased at first, then deceased for several of the tested AV levels. This was mainly due to the gradual decrease in exergy efficiency during the drying process. The grain and exhaust temperature increased, and the exergy rate of the system outlet gradually increased; thus, the IPR gradually increased as well. When the drying process reached a certain point, the grain and exhaust temperature reached a stable level, and the increase was slow or nonexistent; the improvement potential reached its maximal value, then gradually decreased. Therefore, the process parameters need to be taken into careful consideration in order to better improve equipment and energy efficiency when drying to a stable state. When the AV was 0.33–0.43 and 0.47–0.5 m/s, the IPR value of the drying process was between 1.9 and 5.6 J/s, and 5.2 and 9.5 J/s, respectively; the higher the AV was, the greater the IPR. This is the same as the research results of Beigi [23], which showed that the exergetic IPR of deep bed drying of rough rice in a convective dryer increased with the increase in flow rate.

As shown in Figure 13, the SI first decreased, then increased with the drying process at AV of 0.33, 0.37, and 0.4 m/s, ranging from 2.6 to 6.1. This fluctuated slightly at an AV of 0.43 m/s, ranging from 2.1 to 3.0. The trend of change was relatively flat, in the range of 1.6–2.3. With the increase in AV, the SI of drying process decreased, which is the same as Beigi [23]’s research results, which showed that the SI of deep bed drying of rough rice in a convective dryer increased with the reduction in the drying air flow rate.

The above analysis shows that the IPR of the drying process was lower and the SI was higher when the drying AV through the grain layer was less than 0.43 m/s.

## 4. Conclusions

Drying is an operation that involves high energy consumption and pollution. Thus, energy saving and environmental protection are goals of the drying industry. Exergy analysis of the drying process and seeking better drying process parameters are ways to solve challenges involving energy saving and environmental protection in drying. In this paper, a design for an electric stationary bed drying apparatus with ICODM was developed, and a rice drying experiment was carried out using this apparatus. We analyzed the influences of the two important drying parameters, AT (30–55 °C) and AV (0.33–0.5 m/s), on EEE parameters such as the IPR and the SI of the drying process. The following conclusions were drawn:(1)For drying rice in an electric stationary bed with ICODM, when the AT was between 30 and 50 °C the energy efficiency of the drying process gradually increased, while exergy efficiency did not significantly change. When the AT was between 50 and 55 °C, the energy efficiency of the drying process barely changed, while the exergy efficiency gradually decreased. When the AT was between 30 and 55 °C, AV across the grain layer was 0.5 m/s, and the average EEE of the drying process was 48.27–72.17% and 40.27–71.07%, respectively, and increased with the drying temperature.(2)When the AV through the grain layer was 0.33–0.5 m/s and the AT was 40 °C, the EEE of the drying process first decreased and then increased; the average values were 39.79–73.9% and 49.66–71.04%, respectively, and decreased with the increase in AV.(3)When the AT was between 30 and 50 °C, the IPR of the drying process decreased gradually to between 4.1 and 8.5 J/s; however, the SI of the drying process did not significantly change, remaining between 1.9 and 2.7.(4)When the AV through the grain layer was 0.33–0.5 m/s, the IPR of the drying process first increased and then decreased, but the SI of the drying process first decreased and then increased. When the AV through the grain layer was lower than 0.43 m/s, the IPR of the drying process was low, while the SI was high.

In this paper, the effects of single factors on drying temperature and AV through the grain layer on exergy characteristics are discussed. The effect of the interaction between these two factors on exergy characteristics requires further study based on orthogonal experiments. Then, a mathematical model of the effect of drying parameters on exergy characteristic can be established and the process control parameters refined from the fine drying process, obtaining the best energy-saving effect.

## 5. Patents

Condensation heating grain drying basic test device, Chinese patent: 212030056U, 2019.

## Figures and Tables

**Figure 1 foods-11-00101-f001:**
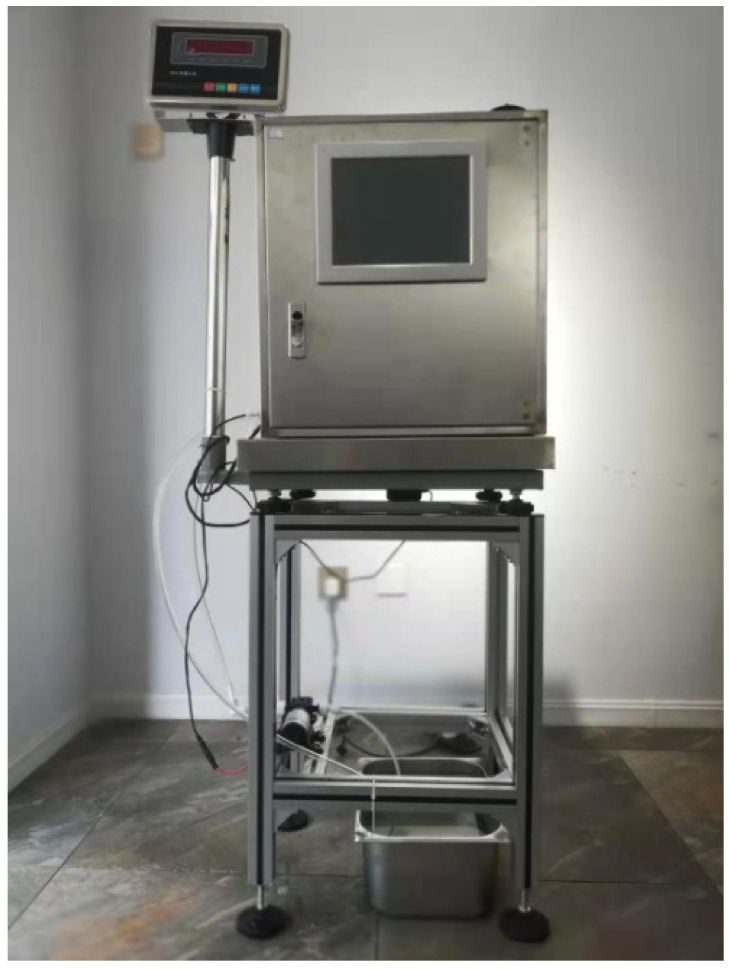
Experimental apparatus photo.

**Figure 2 foods-11-00101-f002:**
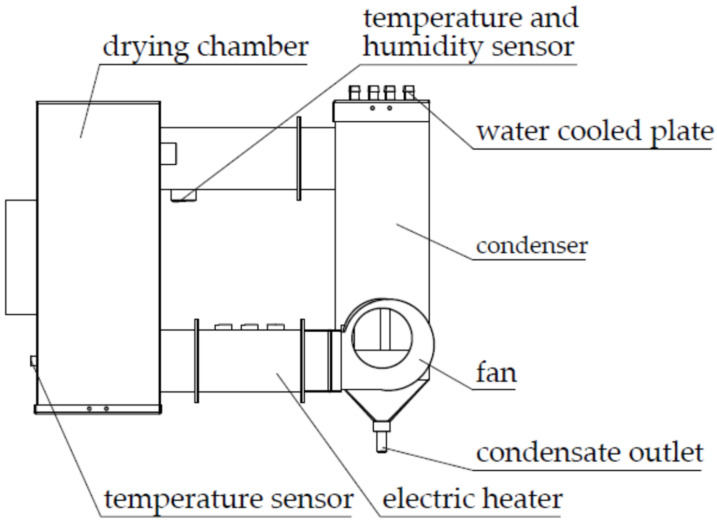
Design drawing of experimental apparatus.

**Figure 3 foods-11-00101-f003:**
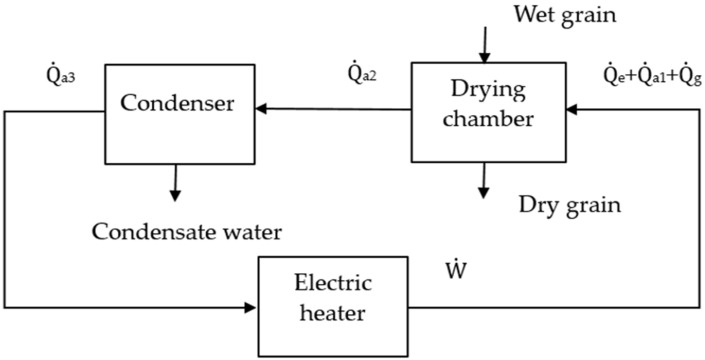
Energy flow diagram of the experimental apparatus.

**Figure 4 foods-11-00101-f004:**
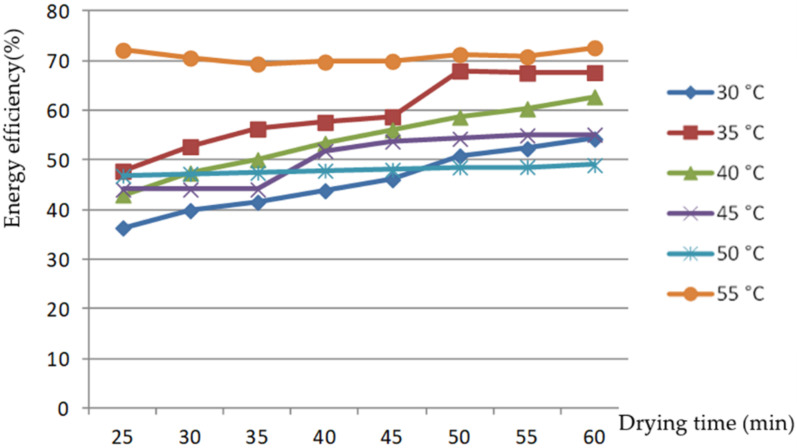
Changes in energy efficiency with drying time at different air temperatures.

**Figure 5 foods-11-00101-f005:**
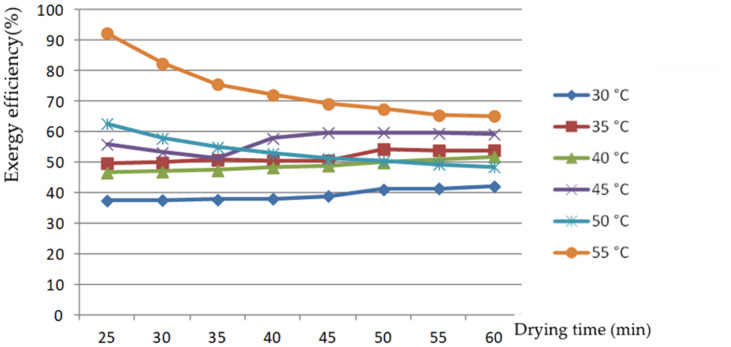
Changes in exergetic efficiency with drying time at different air temperatures.

**Figure 6 foods-11-00101-f006:**
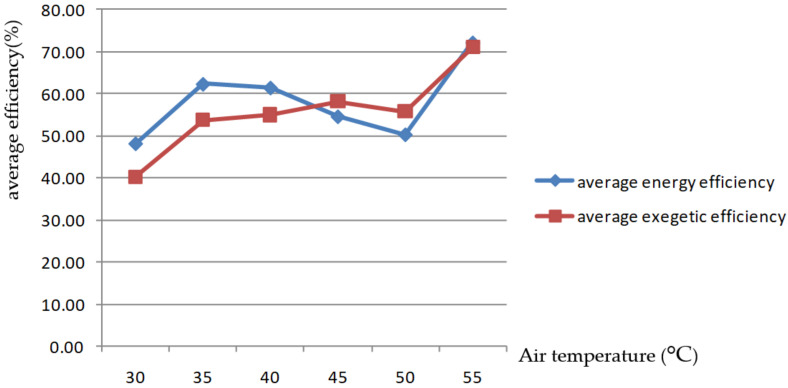
Changes in average energy efficiency and exergy efficiency of the drying process with temperature of drying medium.

**Figure 7 foods-11-00101-f007:**
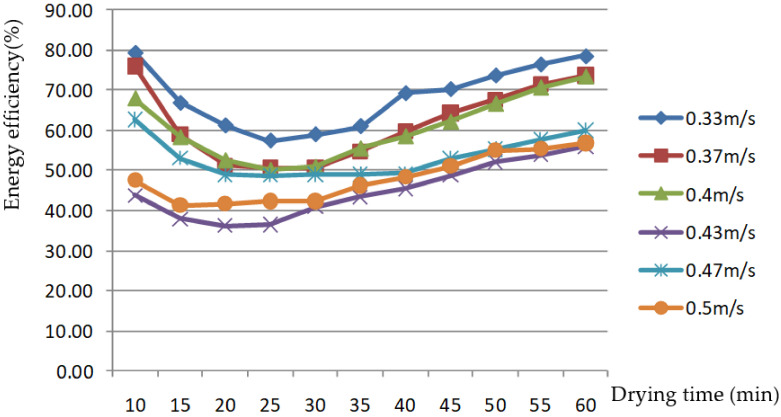
Energy efficiency changes with drying time at different air velocities.

**Figure 8 foods-11-00101-f008:**
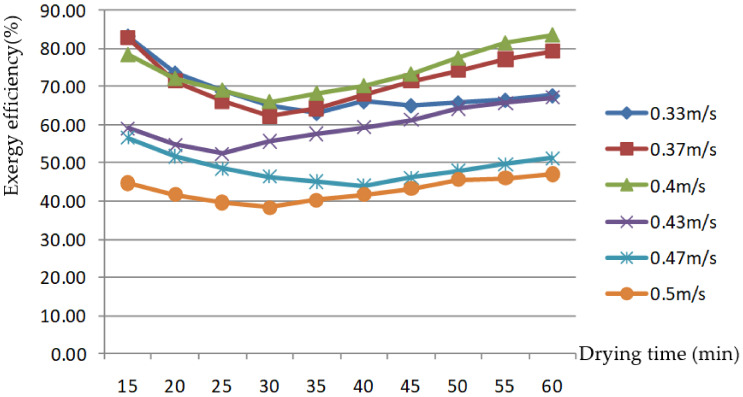
Exergy efficiency changes with drying time at different air velocities.

**Figure 9 foods-11-00101-f009:**
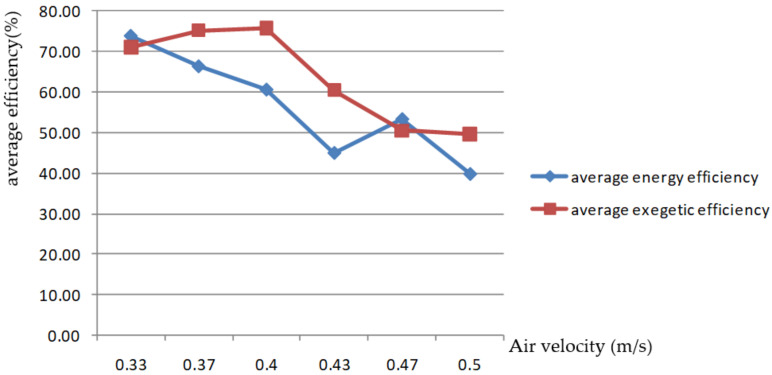
Changes in the average energy efficiency and exergy efficiency of the drying process with flow velocity of the drying medium.

**Figure 10 foods-11-00101-f010:**
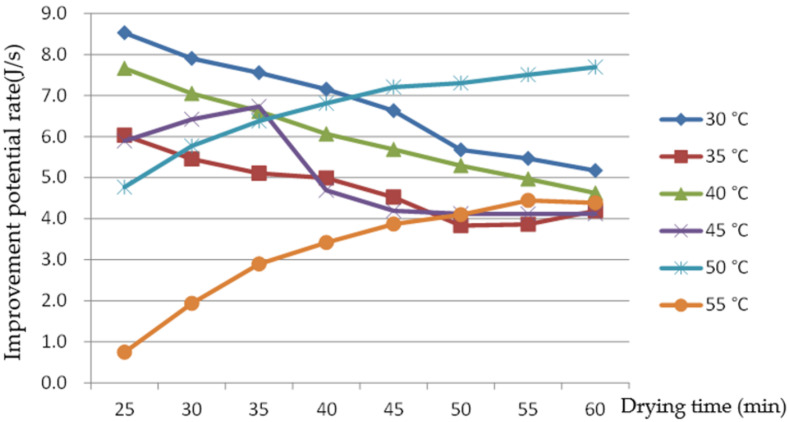
The improvement potential rate changes with drying time at different air temperatures.

**Figure 11 foods-11-00101-f011:**
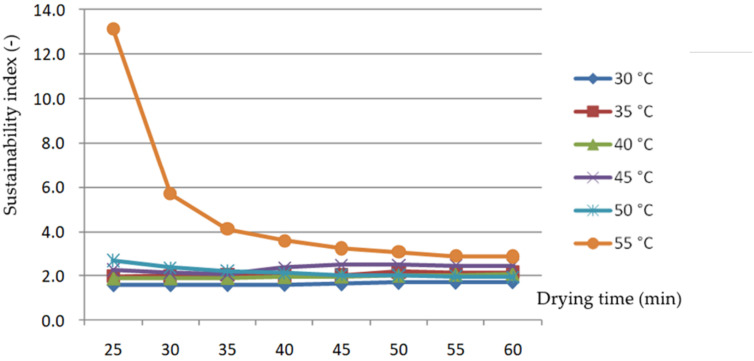
Changes in the sustainability index changes with drying time at different air temperatures.

**Figure 12 foods-11-00101-f012:**
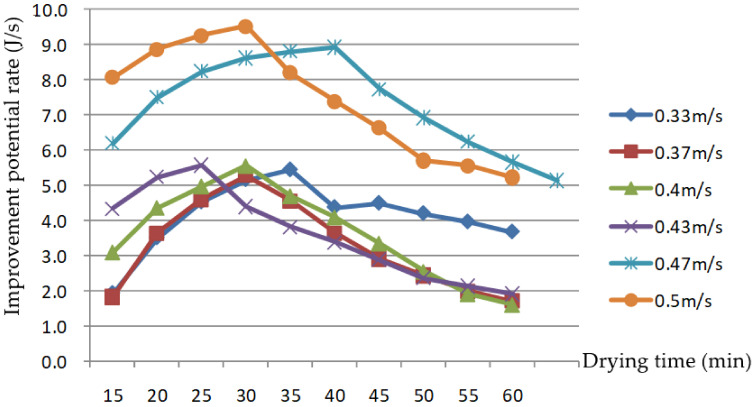
Changes in the improvement potential rate with drying time at different air velocities.

**Figure 13 foods-11-00101-f013:**
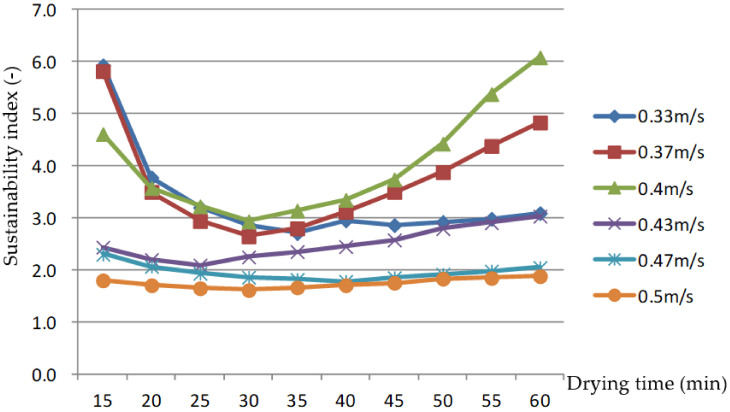
The changes in sustainability index rate with drying time at different air velocities.

**Table 1 foods-11-00101-t001:** Studies on the effect of drying process variables on exergy efficiency in the drying process and their conclusions.

Product	Drying Type	Process Variable	Conclusion	Reference
Rough rice	Laboratory-scale plug flow fluidized bed dryer	Inlet drying AT; feed mass flow rate; weir height	Exergy efficiency of drying chamber varied between 64.39 and 85.45% and increased with increase in drying AT and decrease in both feed mass flow rate and weir height.	[22]
Rough rice	Laboratory-scale convective dryer	Drying AT; AV	Exergy efficiency of drying chamber increased following the increase in both the drying AT and flow rate.	[23]
Kodo millet grains and fenugreek seeds	Wall heated fluidized bed dryer	Wall temperature; AV; bed height	Exergy efficiency increased with increasing wall temperature, AV, decreased with increasing bed height	[24]
Paddy	Vertical fluidized bed dryer	Drying AT; mass of paddy	Drying AV was 2 m/s; exergetic efficiency increased with increasing drying AT, while the converse was the case at 3 m/s	[25]
Paddy	Industrial fluidized bed	Drying AT; initial moisture content	During drying of a higher initial moisture content paddy of 23.66 ± 0.96%wb, energy usage and energy usage ratio increased and exergy efficiency decreased with higher drying AT of 116–125 °C when compared with drying at a comparatively lower initial moisture content (22.04 ± 0.35%wb) and using a lower drying temperature of 103–113 °C	[26]

**Table 2 foods-11-00101-t002:** Models and parameters of the main components.

Name	Model	Parameters Specification
Control of the host	PPC-DL104D	10.4-inch industrial all-in-one machine
Aluminum water cooler	40 × 240 mm	Thickness 12 mm
fan	BFB1012H	Air flow 0.712 m^3^/min, air pressure 249.082 Pa, motor speed 3600 rpm
Electric heating wire	24v25w	_(There are no detailed specifications)

**Table 3 foods-11-00101-t003:** Detection component model and parameter description.

Detecting Parameter	Model	Range	Precision
Air temperature	PT100	−100–280 °C	0.1 °C
Temperature and humidity before and after exhaust condensation	HC2A-S	Humidity range: 0~100% RH Temperature range: −50~100 °C	±0.8% RH/0.01 °C
Moisture content	PM-8188-A	8.0–35.0%	0.5%
weight	YHC	30 kg	1 g
Power meter	DDSU666	_(There are no detailed specifications)	0.001 kwh

## Data Availability

The datasets generated for this study are available on request to the corresponding author.

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
