# Peer review of "Energy and Exergy Analyses of Rice Drying in a Novel Electric Stationary Bed Grain-Drying System with Internal Circulation of the Drying Medium"

_foods, 2021, doi:10.3390/foods11010101_

Round 1
Reviewer 1 Report
The novelty or contrition of the paper should be clearly identified. For example, similar works on exergy analysis have been done but the authors did not mention in the manuscript. Some example are below:
- Exergetic performance assessment of plug flow fluidised bed drying process of rough rice, January 2013,International Journal of Exergy 13(3):387 – 408 DOI: 10.1504/IJEX.2013.057357
- Energy, exergy, and milling performance of parboiled paddy: an industrial LSU dryer,,MA Islam, MHT Mondal, M Akhtaruzzaman… - Drying …, 2021 - Taylor & Francis
- Energy and exergy analyses of paddy drying process in a vertical fluidised bed dryer, January 2019International Journal of Exergy 28(2):113 DOI: 10.1504/IJEX.2019.097975
Line 15- could be EEE or 3E instead of EAEE
Abstract should contain a one line summary results of IPA and SI results
Line 35- in Given – use small G
Line 43- did you mean pumps in “heat bumps”?
Citation should be corrected throughout – for example in line 63 and 65, the formate should be XX etc al [num].
Materials – it’s a very small scale experiment with only 2 kg rice, 340g of sample? Is the analysis going to be same for bigger scale ?
Line 114- what is lagging layer of airflow?
Line 120- what do you mean by quality here, quality will take different definition based on products and use.
Line 179-183- reduce spacing
Line 195- put reference of equations – this has been used by other authors
Fig references should be consistent e.g. line 254 and 284
Should be a space in Fig XXX
What IPR physically mean ?
Author Response
Dear reviewer,
I am very grateful to your comments for the manuscript, according to your suggestion, I have made detailed revised to the manuscript, please see the
attachment for details.

Reviewer 2 Report
This study is very interesting because a good drying system was developed in which energy consumption and pollutant discharge were reduced in the process. An innovative technology for drying grains in a stationary bed was presented based on the internal circulation of the drying médium.

Author Response
Dear reviewer,
I am very grateful to your comments for the manuscript, according to your suggestion, I have made detailed revised to the manuscript, please see the attachment and revised manuscript for details.

This manuscript is a resubmission of an earlier submission. The following is a list of the peer review reports and author responses from that submission.